

# Exploration of virtual catchments approach for runoff predictions of ungauged catchments

Jun Zhang[1], Dawei Han[1], Yang Song[1], Qiang Dai[2]

[1]Department of Civil Engineering, University of Bristol, Bristol BS8 1TR, UK

[2]Key Laboratory of VGE of Ministry of Education, Nanjing Normal University, Nanjing, 210000, China

*Correspondence to*: Jun Zhang (jun.zhang@bristol.ac.uk, catherine.zjun@gmail.com)

*Competing interests.* The authors declare that they have no conflict of interest.

**Abstract.** We explore unit hydrograph (UH) properties influenced by catchment geomorphology that could be used in ungauged catchments. Unlike using gauged catchments, a robust approach with virtual catchments was adopted in deriving UH equations. Over 2000 virtual catchments were created from the baseline model of the Brue catchment, UK. A distributed model, SHETRAN, was used to generate runoff in these catchments. Using virtual catchments is feasible to control catchment geomorphologies, which could not be done with the real catchments due to their vast heterogeneity. Catchment characteristics of average slope, drainage length and a new index of catchment shape were examined of their influence on UH properties. The agreement of the results with the hydrological principles is a useful validation of the approach (e.g., the increasing slope led to quick response to peak ($Tp$) and high peak volume ($Qp$) of UH, whereas the drainage length presented an opposite trend). Catchment shape was shown to have a significant effect on UH properties. Compared with the widely used empirical equation from the U.K. Institute of Hydrology, the drawn conclusion recommends more indicators to be included to derive more comprehensive equations: apart from catchment geomorphologic properties, storm patterns including storm intensity and temporal distribution are also influential on the UH shape. The indicators in this study were limited to generate a sophisticated equation for use. However, these results can be considered as a testing case to gain more understanding in hydrologic processes for ungauged catchments with the help of the virtual catchment approach.

## 1 Introduction

Runoff modeling in ungauged catchments needs a large quantity of data for purposes of generalization. Our knowledge of catchment responses is not adequate to simply transfer models derived from a gauged to an ungauged catchment (Sivapalan, 2003). The heterogeneity of catchment geomorphology, e.g. its terrain, area, shape, land surface condition, soil types, etc., is the root cause of the difficulty in predicting catchment response (Hrachowitz et al., 2013; Pilgrim et al., 1982; Sivapalan, 2003). Transferring model parameters from one catchment to another is linked to regional catchment characteristics (Bárdossy, 2006; Castiglioni et al., 2010; Young, 2006) as the dominant control on runoff production and routing (Beven et al., 1988; Beven and Wood, 1983). Therefore, understanding the catchment geomorphological impact on runoff is crucial to predicting streamflow in ungauged catchments. Among all hydrological approaches, the unit hydrograph (UH) is recognized as being an effective prediction tool and is deemed to reflect the characteristics of the catchment with the potential to estimate the streamflow in ungauged catchments (Sherman, 1932).

Hydrologists have been attempting to derive UHs from catchment descriptors for decades. The possibility to extract UHs from catchment characteristics was proposed by Bernard (1935), followed by the early synthetic UH development strategies (Snyder, 1938; Taylor and Schwarz, 1952), most of which are empirical methods. Further examples of empirical methods can be seen in U.S. Soil Conservation Service (Mockus, 1957) and Singh (1988). The U.K. Institute of Hydrology offered an empirical UH that used 1822 individual rainfall runoff events and 204 catchments in the country (Robson and Reed, 1999). Catchment area, drainage length, distance to the outlet, land use, antecedent soil condition etc., are considered in these methods. Most





traditional UH methods establish a set of empirical relations among catchment characteristics to describe the shape of UH on
the basis of gauged catchments, which have certain region-specific constants/coefficients varying over a wide range (Singh et
al., 2014). Its inconsistency due to subjectivity and manual fitting makes it a challenge to apply in ungauged catchments or
different regions. Moreover, another main obstacle is how to choose catchment data sets properly. Either small data sets or
catchments with widely varying characteristics hamper the derivation of a UH that can be used to represent the ungauged
catchments (Van Esse et al., 2013; Robinson et al., 1995).
Apart from empirical methods, conceptual models represent a catchment as a series of linear storages and are based on
continuity equations and the storage discharge (Clark, 1945; Nash, 1957). This simplification ignores the flow translation in
the catchment, which is essential to describe the behavior of a dynamic system. What is more, the coefficients are difficult to
determine  for ungauged catchments; for example, some parameters should ideally be an integer rather than a fractional value
derived from catchment characteristics (Singh et al., 2014). With the increasing availability of geographic information system
(GIS) data, GIS-supported UH approaches are explored by Jain et al. (2000), Jain and Sinha (2003), Sahoo et al. (2006), and
Kumar et al. (2007). These approaches are based upon conceptual UH models to evaluate model parameters related to
geomorphological characteristics from GIS packages. However, this improvement of modeling performance does not solve
the problems brought by conceptual simplification of the models.
Explicitly integrating catchment geomorphology details in the framework of travel time distribution to define a
geomorphological instantaneous unit hydrograph (GIUH) is pioneering work that is appeared to be promising in ungauged
catchment modeling (Rodríguez-Iturbe and Valdes, 1979; Rodríguez-Iturbe et al., 1982; Valdés et al., 1979). The model
properties assessed are highly dependent on geomorphologic elements (Chutha and Dooge, 1990). The GIUH model has
generated a wealth of research since its conception. However, it has been criticized because of its assumption of exponential
distribution of drainage time mechanism by Gupta and Waymire (1983), Kirshen and Bras (1983), and Rinaldo et al., (1991),
and the assumption of uniform celerity of flow has been shown to actually change from storm to storm (Pilgrim, 1977).
Moreover, GIUH is not sensitive to rainfall spatial distribution due to the averaging scheme used in the model (Corradini and
Singh, 1985). Further, ignoring the effect of catchment slope is unreasonable especially in because small catchments, which
are sensitive to hillslope response (Botter and Rinaldo, 2003; D'Odorico and Rigon, 2003; Robinson et al., 1995).
The width function-based GIUH (WFIUH) model employed by Rinaldo et al. (1995) shows a better capability to model
different transport processes within channeled and hillslope regions. WFIUH has been further improved by Grimaldi et al.
(2010, 2012) with spatially distributed flow velocity within a digital elevation model (DEM)-based algorithm. Nevertheless,
there are still practical limitations in extracting the network widths especially for large catchments (Sahoo et al., 2006). The
freezing parameters for the whole catchment limit the variability of the hydrological scheme (Rigon et al., 2016).
Previous studies have put great effort into looking for a general rational formulation linking UH properties with catchment
geomorphology. However, due to the limitation of data availability and conceptual simplification of hydrological processes,
the lack of understanding of physical principles hinders parameter transferring from gauged to ungauged catchments. To
overcome the aforementioned issues, we present a series of virtual catchments to explore streamflow generation in ungauged
catchments. Each virtual catchment is assigned particular catchment characteristics including three main indicators i.e., average
slope and drainage length, and catchment shape, which are rarely accounted in previous studies. The experiments are
undertaken using a fully distributed model, Système Hydrologique Européen TRANsport(SHETRAN),  for simulating the
catchment response, the results of which are compared with the standard UH equations in the widely used Flood Estimation
Handbook (FEH) proposed by the U.K. Institute of Hydrology in UK (Robson and Reed, 1999).



**2    Methodology and data sources**
**2.1    SHETRAN**
SHETRAN is a physically based spatially distributed hydrological model for water flow and sediment and solute transports in
catchments (Ewen et al., 2000), which is originated from the Système Hydrologique Européen (SHE) (Abbott et al., 1986).
SHETRAN provides an integrated representation of water movements through a catchment, containing major elements of the
hydrological cycle as shown in Table 1. It models streamflow in a single complete river catchment by retrieving data for a
catchment, including weather data, river gauge recordings, catchment properties, e.g. DEM, land use and soil type. The
catchment is represented by an orthogonal grid, which allows spatial distribution of input data, including rainfall, metrological
data and catchment properties, etc. The model has been applied in varied catchments and has proved to be a reliable
hydrological model (Birkinshaw and Ewen, 2000; Hipt et al., 2017; Norouzi Banis et al., 2004; Zhang et al., 2013).
**2.2    Flood Estimation Handbook prediction**
An empirical equation of instantaneous unit hydrograph (IUH) derived from regression analysis with 1822 individual events
and 204 catchments has been proposed by the U.K. Institute of Hydrology and has been widely adopted in practical streamflow
prediction; it is referred to the FEH equation in this study (Robson and Reed, 1999). Four catchment descriptors are used to
predict the time to peak (Tp) in IUH as shown in Equation (1).

93        $Tp = 4.270 \, S^{-0.35} \, W^{-0.80} \, L^{0.54} \, (1-U)^{-5.77}$                      (1)

in which, $Tp$ is the time to peak, in h; $S$ is the mean drainage path slope (m km$^{-1}$); $W$ is the proportion of time when soil
moisture deficit is below 6mm (%); $L$ is an index describing drainage path (km); $U$ is the extent of urban and suburban land
cover (km$^2$).
At each grid node, an outflow direction is defined to one of its eight neighboring nodes. Using the difference in altitude and
the distance between the two nodes, the internode slope is calculated. The procedure is repeated for all nodal pairs within the
catchment to give the mean drainage path slope $S$. Using the drainage paths, the distance between each node and the catchment
outlet along the flow path is calculated; $L$ is the mean value of all these distances. To compare the results from virtual
catchments with the FEH equation, the same methodologies are applied in the following analysis when referring to the same
factors.
In the FEH equation, the IUH peak volume $Qp$ is derived from $Tp$ as a regression result and a continuity constraint as shown
in Equation (2)(Robson and Reed, 1999).

105        $Qp = \frac{2.2}{Tp}$                                       (2)

in which, $Qp$ is the peak volume of UH (mm).
In practice, using an optimal interval UH gives a much smoother response. Therefore, it is customary to use convenient
values such as 0.25, 0.5 or 1 h in UH, which can be done with Equation (3)(Robson and Reed, 1999).

109        $Tp(\Delta t) = Tp(0) + \frac{\Delta t}{2}$                            (3)

**2.3    Study area**
This study explores the rainfall-runoff behavior embedded with changing catchment geomorphology in the Brue catchment,
UK. The Brue catchment has been a focus of research because of the abundant available data (Dai and Han, 2014; Dai et al.,
2014, 2015; Younger et al., 2009). The Brue catchment comprises 137 km$^2$ of the river's headwaters and drains to the river
gauge in Lovington (Moore et al., 2000). Figure 1 presents the general spatial characteristics of the catchment in a 500m size
grid that is used in the following analysis. The elevation varies from 251 m in the northeast to 22 m in the southwest. There





are three soil types, i.e. mud, clay and sand, with their distribution shown in Fig. 1 according to the national soil type data
downloaded from Digimap Service (Soil Parent Material Model, 2011). The slope ranges from 10.85% to 0.07% (transferred
to m km$^{-1}$ for further analysis) and shares a similar spatial patterns with the elevation. Flow length mainly depends on the
distance from the node to the outlet, ranging from 0 up to 19.81 km.
Prior to the execution of the virtual catchments, a baseline distributed model of the Brue catchment was configured and then
calibrated and validated with historical measured data in the Brue catchment. The calibration was based on the hourly data in
1995 and validation with the hourly data in 1996 involved the collection of discharge and meteorological data. We used the
streamflow at the outlet for model calibration and validation. The baseline model of the Brue catchment was constructed from
a 50 m DEM with the grid cell size of 500 m. The stream network was derived automatically from the DEM in the model. The
average slope of the catchment is 29.18 m km$^{-1}$ and the average drainage length is 12.12 km.
The catchment was schematized as an orthogonal grid in SHETRAN integrated with a spatially variable geomorphology. To
examine the runoff response to variable catchment characteristics, change of catchment geomorphologies (mainly on the role
of average slope, drainage length and catchment shape) were specified in a large number of virtual catchments. When verifying
one element, the other geomorphological features were kept unchanged. A UH was generated from each simulation to evaluate
the relationship between UH and catchment properties, which were eventually used in ungauged catchments. During the
experiment, although some of the virtual catchments were extreme when compared with the real catchment conditions, they
still allowed the useful insights to be gained in understanding the role of catchment geomorphology on runoff generation.

## 3    Results

### 3.1    Model validation

With the real soil map information, we used experimental soil parameters by the Boreal Ecosystem-Atmosphere Study Data
Sets hydrology (BOREAS HYD-01) team (Kelly and Cuenca, 1998). However, due to a lack of land use data in the catchment,
a homogeneous land use map was assigned in the model with calibrated land use parameters. The model was evaluated with
Nash-Sutcliffe efficiency (NSE) (Nash and Sutcliffe, 1970), which is generally adopted in hydrological research (Guerrero et
al., 2013; Parasuraman and Elshorbagy, 2008; Rojas-Serna et al., 2016; Zhuo et al., 2015).
In previous hydrological modeling, the simulation of discharge is commonly acceptable when NSE is greater than 0.8 (Beven
and Binley, 1992; Freer et al., 1996). In this study, the calibrated NSE was 0.82 and the validated NSE 0.81 with the hydrograph
shown in Fig. 2. Due to the calculation function of NSE, it is more sensitive to higher flow than lower values so the analysis
is very useful in peak flow studies, as shown in Fig. 2 (Krause et al., 2005). There are numerous rainfall events in both dry and
wet seasons of one year in the Brue catchment, therefore, the model is fully excited with abundant information. The qualified
performance demonstrates that SHETRAN is capable of providing a realistic representation of the catchment hydrology in the
case site.

### 3.2    Average slope

Response of streamflow on the average slope was examined by changing the elevations across the catchment on the basis of
the original topography. All the grids were multiplied by the same factor varying from 0.2 to 4 in individual virtual catchments
so that the average slope changes from 5.84 m km$^{-1}$ to 116.72 m km$^{-1}$. In the meantime, other properties such as catchment
area and grid size remained unchanged to avoid compounding effects. A uniformly distributed rainfall of 10 mm for 1 h was
applied for simulating outlet runoff in SHETRAN.
With the slope varying from 5.84 m km$^{-1}$ to 116.72 m km$^{-1}$, $Tp$ in the UH decreased from 17 to 7 h with a clear power
declination displayed in Fig. 3(Kirpich, 1940; Robson and Reed, 1999). Since the catchment is relatively small, the runoff





generation is not quite sensitive in the hourly time step, which presents the same values among similar slopes, e.g. the slope
group from 70 m km⁻¹ to 116.72 m km⁻¹.
By testing several trend lines with the results and based on the reference of FEH equations, a power function is used in this
study to describe the trend. A standard form of power function $y = ax^b$ was integrated with two coefficients $a$ and $b$. To
distinguish from the other coefficients, the above were labelled $a_1$ and $b_1$ in this experiment and specific subscripts are used
respectively for the following results. Compared to the FEH equation stated in Equation (1) with the relationship of average
slope, $b_1$ was -0.35 while $b_1$ was -0.29 displayed in Fig. 3. When the absolute value $b_1$ was smaller, a less significant
relationship was found between the two variables.

### 163    3.3    Drainage length

#### 164    3.3.1    Drainage length and average slope

The same uniform rainfall as the previous experiment was distributed for the experiment on the effect of drainage length ($L$)
on runoff response. The catchment cell size was multiplied by certain factors ranging from 0.1 to 10, leading to values of $L$
from 1.21 km to 121.20 km. Nine different groups of virtual catchments were generated with different slope values for further
analysis, ranging from 14.59 to 116.72 m km⁻¹. To maintain the average slope of these virtual catchments unchanged in each
group, the elevation values in each catchment were also changed by multiplying them by the same factor. In Fig. 4, five of the
nine groups are marked in the legend with the multiple factor of the slope, e.g., 'slope 0.5' means the average slope of this
groups is 14.59 m km⁻¹. The derived equations are listed in Table 2 with $a_2$ and $b_2$ for the nine groups.
All groups demonstrated a similar trend in which the longer $L$ is, more time it needed to reach the peak volume of UH, which
presents a power function as well. Moreover, when the streamflow drained the same length with different slopes, it took a
shorter time to reach the outlet on steeper catchments, which is consistent with the result in the previous sections and other
studies (Kirpich, 1940; Robson and Reed, 1999).
As seen in Table 2, $a_2$ decreased from 0.94 to 0.53 when the slope increased from 14.59 to 116.72 m km⁻¹. Meanwhile, $b_2$
experienced a slight fluctuant around 1.10. The coefficient $a_2$ is best represented at the starting point of the line, while larger
$a_2$ indicated a longer $Tp$ when the catchment is small and flat. Referring to the derived equations, both $a_2$ and $b_2$ decrease in
steeper catchments. Values of $b_2$ are greater than that adopted in the FEH equation (0.54), which presents a greater rate of
increase of the trend. The results demonstrate that $Tp$ from the catchments in the experiments are more sensitive to $L$ than in
the FEH equation. Moreover, $b_2$ experienced an increase with the increase of slope followed by a decrease, which implies that
$L$ is of more importance when the catchment is steeper. However, the significance of $L$ was weaker when the steepness kept
increasing.

#### 184    3.3.2    Drainage length and storm patterns

The experiments on changing slope and $L$ showed similar trends with the FEH equation. However, the coefficients were vastly
different. In the previous experiments, a homogenous rainfall with 10 mm in 1 h was applied, which is not realistic for real
catchments and also different from the storms chosen to derive the FEH equation. It is also shown that UHs from varied storms
can be different (Corradini and Singh, 1985; Rigon et al., 2016; Valdés et al., 1979). Thus, to further explore the UH generation
from storms with different intensities and durations, multiple rainfall events were applied to the virtual catchments.
$S$, $W$ and $U$ were held constant at the original value of the catchment and only $L$ was changed in this experiment. FEH was
derived by replacing $W$ and $U$ with the real data and eventually presented by a function with an independent variable $L$.
The relationship between $Tp$ and $L$ derived from varied storms is shown in Fig. 5 and Table 2 with coefficients of $a_3$ and $b_3$.
Table 2 presents a full list of experiments with examples of experiments shown in Fig. 5. In Fig. 5 (a), rainfall is uniformly
distributed in the catchment with varied intensities (from 10 mm to 80 mm) for 1 h. A clear trend was seen in each storm





between $Tp$ and $L$, while for larger storms both $a_3$ and $b_3$ decreased, as seen in Table 2. $Tp$ appears to decrease for larger
storms in both small and large catchments. Moreover, the difference between small and large catchment becomes smaller when
the storm intensity increases. The larger $a_3$ in the FEH equation was overestimated in small catchments while the smaller $b_3$
illustrates that underestimated $L$ influences discharge predictions for large catchments.
More patterns are presented in Fig. 5 (b) when storms with different durations of the same rainfall intensities (10 mm h$^{-1}$) were
explored. When the duration increased, $a_3$ presented an increasing trend while $b_3$ showed an opposite trend. Similar to what
was seen in Fig. 5 (a), large storms were less sensitive to catchment size with decreasing $b_3$. However, increasing $a_3$ illustrates
that small catchments took longer to reach the peak volume when the storm duration increased. For catchments with longer
drainage length, there was a lower effect of storm duration on runoff generation, which also exhibited a declining trend.
More comparisons were carried out between different temporal distributions of rainfall on runoff generation as shown in Fig.
6 and Table 2. Figure 6 (a) presents $Tp$ derived from 10 mm, 20 mm and 50 mm in varied durations. For the storms of 20 mm
in two temporal patterns, $Tp$ showed little difference with similar values between coefficients of $a$ and $b$. However, the storm
of 50 mm displayed a more apparent trend of $Tp$, with $L$ in two patterns. Larger discrepancies can be found in Fig. 6 (b), which
depicts that the storms of 100 mm and 200 mm were more sensitive on temporal patterns. The value of $a$ increases when storm
duration is lengthened while $b$ experienced an opposite trend. When storm duration was longer, it took longer to reach the
peak volume in small catchments. However, if the drainage length was long enough compared to the storm duration, the
influence of storm duration decreased, which is clearly shown in Fig. 6 (a) for the 50 mm storm with an intersection of two
lines. A longer storm required a longer $L$ to eliminate the impact of rainfall duration, as displayed in the lines for 100 mm and
200 mm in Fig. 6 (b).

### 214    3.4    Catchment shape

Catchment shape is not among the factors included in the FEH equation as well as other research on catchment geomorphology.
A simple experiment with three catchments of different shapes was carried out in this study as shown in Fig. 7 with the general
information in Table 3. Catchment A is the original Brue catchment, and the other two catchments are its transformed clones.
Catchment B was transformed by extending the catchment in the north-south direction and shortening in the east-west direction
of the original catchment, while Catchment C is in the other way around, i.e., lengthened E-W and shortened N-S. Owing to
the symmetry of the original catchment, the generated catchments all had a close resemblance in areas, drainage lengths and
slope. All catchments are represented with the same cell size of 500 m. A homogenous rainfall of 10 mm with 1 h duration
was applied to the three catchments and then the corresponding UHs from varied $L$ was evaluated.
Figure 8 and Table 2 with coefficients of $a_4$ and $b_4$ illustrate the relationship between $Tp$ and $L$ in the three different shapes.
Figure 8(b) is a zoomed-in view of a portion of Fig. 8(a) when $L$ is smaller than 20 km. $Tp$ varied in the catchments with
different shapes even with similar catchment descriptors. With all the other characteristics controlled, the shape of Catchment
C presented the longest drainage time. Catchment A experienced the quickest drainage time when $L$ was greater than 20 km,
and Catchment B was the fastest when $L$ was shorter than 20 km. Focusing on the scatter dots when $L$ is greater than 100 km
in Fig. 8 (a), it is found that $Tp$ from Catchment B and C were close and $Tp$ from Catchment A remained smaller than from B
and C. Moreover, the smallest coefficient in the equation $b_4$ of Catchment A demonstrates that the original shape was the least
sensitive to the change of  $L$.

### 231    3.5    Relationship between $Qp$ and $Tp$

The analysis between $Qp$ and $L$ in terms of different slopes and shapes is displayed in Fig. 9 and Table 4 with the coefficients
of $a_5$ and $b_5$. A power decreasing relationship was also recognized in Fig. 9 for all the experiments, which is also demonstrated
in previous studies (Rinaldo and Rodriguez-iturbe, 1996; Rodríguez-Iturbe and Valdes, 1979). The increased slope results in
higher peak volume while coefficient $a_5$ in the power function was more sensitive to slope than $b_5$ in Fig. 9(a). This means





that the effect of slope and drainage length is more significant for small catchments than for large ones. Moreover, Catchment
C presented the highest peak volume while Catchment B experiences the lowest peak volume as shown in Fig. 9(b).
It is acknowledged from Eq. (2) of the FEH equation that $Qp$ is inversely proportional to $Tp$ with a coefficient of 2.2 for all
the catchments. Similar results can be found in previous studies with different coefficients (Moussa, 2003; Rinaldo and
Rodriguez-iturbe, 1996). Taking $y = c/x$ as a representation of the inversely proportional function with $c$ as the main
coefficient, $Qp$ from some previous experiments are plotted against $Tp$ as shown in Fig. 9(c)-(d), with the equations shown in
Table 4. Obviously, an inversely proportional relationship was clearly shown between $Qp$ and $Tp$ for all the experiments.
However, $c$ was not consistent for varied storms in catchments with different geomorphologies. Figure 9(c) and (e) exhibit $Qp$
from storms with varied intensities and temporal distributions. From Fig. 9(c) it can be found that $c$ decreases with longer
storm duration and Fig. 9(e) showed that $c$ increased with larger storm intensities. Figure 9(d) demonstrates that catchment
slope had little influence on the relationship between $Qp$ and $Tp$. What is more, catchment shape is also influential on how to
deduce $Qp$ from $Tp$ in UH. The value of $c$ was largest in Catchment C while smallest in Catchment A. Therefore, this may
result in errors when applying a uniform equation to obtain $Qp$ from $Tp$ in the UH. It is therefore highly recommended to
derive various equations based on the catchment geomorphology as well as storm patterns for prediction in ungauged
catchments.
**4    Discussion**
**4.1    The implementation of virtual catchments**
When transferring knowledge from gauged to ungauged catchments, obstacles exist due to the varied geomorphology and
storm types between catchments. Data scarcity also hinders the development of a uniformly acceptable approach for ungauged
catchments (Hrachowitz et al., 2013). A common shortcoming of previous empirical UH derivation is that the catchments used
are extremely diverse (Corradini and Singh, 1985; Robinson et al., 1995; Robson and Reed, 1999; Sawicz et al., 2014; Valdés
et al., 1979). For instance, it is hard to control other characteristics (e.g., drainage length, shape, soil, etc.) when we try to
investigate how a single catchment element (such as average slope) affects the unit hydrograph. It is already pointed out that
error exists with the coefficient of determination $r^2 = 0.74$ from the FEH-derived equation (Robson and Reed, 1999).
Moreover, the existing conceptual UH models are mostly based on Horton ratios or other river disciplines (Chutha and Dooge,
1990; Gupta et al., 1980; Rodríguez-Iturbe and Valdes, 1979), which are prone to ignore some catchment features. Peña et al.
(1999) found that there was not a unique hydrograph for distinct watersheds with similar Horton ratios.
In this study, we have used a virtual experiment approach to seek new understanding of the impacts of catchment
geomorphology on runoff generation. Our baseline catchment is the well-studied Brue catchment, using the widely-used model
SHETRAN, which has been demonstrated to simulate runoff realistically. Using a fully distributed model, we performed a set
of model experiments to simulate the streamflow in thousands of virtual catchments. The experiments explored how catchment
geomorphology could influence the runoff generation in terms of UH properties, which is informative in ungauged catchments.
The advantage of virtual catchments is their simplicity, avoiding unnecessary compounding interferences by controlling of
catchment geomorphology by defining catchments with desired features. As many catchments as required can be created in
this way, solving the problem of data scarcity. With a reliable distributed hydrological model and synthetic rainfall input, the
corresponding streamflow is generated without the required measurements. However, extreme catchments are possible,
inducing unusual streamflow and potential modeling error. It is worthwhile to consider the boundary conditions carefully when
creating catchments. Besides studies of catchment geomorphology, other spatial variability, such as rainfall input, parameter
heterogeneity can also be carried out in virtual catchments. More comprehensive results of runoff generation can be obtained
with the help of the virtual catchment approach, which can provide further useful information for ungauged catchments.



### 4.2 UH properties from different catchment geomorphologies

As widely applied in ungauged catchments, $Tp$ and $Qp$ are the most important properties to determine UH shape. Average slope, drainage length, catchment shape and storm properties were examined in this study. The slope and drainage length were generally considered in the previous studies but not catchment shape. As acknowledged, nearly all catchment shapes are irregular and it is impossible to have two catchments with the same shape in the real world. Therefore, we explored the effect of catchment shape on runoff generation by virtual catchments. Moreover, we compared the relationship between catchment characteristics with $Tp$ and $Qp$ with the FEH equation.

#### 4.2.1 $Tp$ from different catchment geomorphologies

Similar to previous studies, the increase of average slope and drainage area both prolonged the time of flow to reach peak volume with a clear power relationship (Beven and Wood, 1983; Robinson et al., 1995; Robson and Reed, 1999). As noted above, few studies have been done with the regard to the catchment shape. A simple comparison was conducted with three catchment shapes. The results illustrate that the catchment shape was of importance on UH derivation even with similar areas, drainage lengths and average slopes, not to mention the methodologies that used Horton Law to represent the catchment (Chutha and Dooge, 1990; Rodríguez-Iturbe and Valdes, 1979), which defines the catchment morphology with limited indicators. An indicator could be proposed to describe the catchment shape like slope to describe the elevation of the catchment, e.g. the ratio between the distance from the outlet to the farthest node and the longest orthogonal distance. However, quantification of catchment shape requires further investigation.

Nevertheless, compared with a uniform equation applying to all catchments with all storms in the FEH equation, it was found that storm patterns also have an effect on UH generation, which is supported by some previous studies (Corradini and Singh, 1985; Rigon et al., 2016; Valdés et al., 1979). Not only storm intensity but also temporal patterns were crucial for UH derivation. The catchments with longer drainage length are less likely to be influenced by storm duration if the drainage time is greater than storm duration. Moreover, when storm intensity increased with a fixed duration, $Tp$ decreased regardless of how long the drainage was. Deriving possible UHs from the virtual catchments should be more reasonable than simply transferring parameters from gauged to ungauged catchments. More rainfall patterns both in temporal and spatial scales should be investigated for a more complete understanding.

#### 4.2.2 $Qp$ from different catchment geomorphologies

According to the results obtained in the current phase, it has been stated that slope, drainage length and catchment shape all affect peak volume in UH. An inversely proportional relationship is found between $Qp$ and $Tp$ in these results as well as in the FEH approach. Similar to $Tp$, it uses a uniform equation for all catchments and storms in the FEH method. Differences are found in the relationship between $Qp$ and $Tp$ with varied storms and catchment properties. The storm temporal patterns rather than storm intensity were more significant to this relationship. Moreover, the average slope was less important than the catchment shape, but both were much less crucial than the storm distribution. Therefore, it is beneficial to consider storm patterns in deriving $Qp$ from $Tp$ rather than using a uniform equation in FEH.

Overall, increased steepness led to shorter time to peak as well as higher peak volumes, while longer drainage length prolonged the time to peak and brought down the peak volume. Catchment shape, as a new indictor in this study, was also demonstrated to have an influence on UH properties. Moreover, UH was different from storm to storm with varied temporal distribution and rainfall intensities. Therefore, only applying a general UH equation to all catchments and storms does not comprehensively provide precise streamflow prediction in ungauged sites. As a promising and simple approach to be applied in ungauged catchments, UH is worthy of further exploration on how to determine its accurate shape. The virtual catchment approach is a potentially effective method for systematic investigation without the requirement of a large number of gauged catchments.



## 5 Conclusions

UH is widely used in ungauged catchment studies, by deducing equations from gauged catchments. However, it ignores the discrepancy existing between the known with the unknown catchments when transferring parameters. The virtual catchment experiments described here generate a series of relationship between catchment geomorphology and UH properties. Average slope, drainage length and catchment shape (as a new indicator) were shown to be influential on UH properties. Moreover, the UH shape differed from storm to storm caused by the storm duration and intensity. Although $Qp$ has an inversely proportional relationship with $Tp$ as mentioned in the FEH equation, catchment characteristics especially for storm patterns are also crucial to the coefficient used in deducing $Qp$ from $Tp$.

These experiments support the findings in previous studies, and have revealed more catchment characteristics that are influential on runoff generation. In spite of the simplicity of deriving a universal single-UH equation for all catchments and storms, it is feasible to apply a more specific UH to an ungauged catchment considering its geomorphology and the storm characteristics with the help of improved computation technology. More importantly, owing to the virtual catchment approach, a huge number of catchments can be created with desirable features to explore how catchment geomorphology would affect runoff generation. This research suggests an alternative for studying the hydrologic processes in ungauged catchments, especially for regions with data scarcity. However, this study should be regarded as a starting point in obtaining more understanding in hydrologic processes with the help of virtual catchments, and we hope it will encourage more studies in this field to improve the proposed methodology to a higher level.

## 6 Data availability

The atmospheric data and runoff data in the Brue catchment in this study are freely available on the British Atmospheric Data Centre website (https://badc.nerc.ac.uk/home/index.html). The DEM data is freely available on the Digimap website (http://digimap.edina.ac.uk/). The Soil Hydraulic Properties. Data set is freely available online (http://www.daac.ornl.gov) from Oak Ridge National Laboratory Distributed Active Archive Center.

## 7 Acknowledgement

The first author would like to thank the University of Bristol and China Scholarship Council for providing the necessary support and funding for this research. The author Qiang Dai was supported by the National Natural Science Foundation of China (Grant No. 41501429). The authors acknowledge the Dr. Stephen Birkinshaw for the help with SHETRAN in this study.

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





| Processes | Equation |
|---|---|
| Subsurface flow | Variably saturated flow equation (3D) (Parkin, 1996) |
| Overland flow | Saint-Venant equations, diffusion approximation (2D) (Abbott et al., 1986) |
| Channel flow | Saint-Venant equations, diffusion approximation (flow in a network of 1D channels) |
| Canopy interception and drip | Rutter equation (Abbott et al., 1986) |
| Evaporation | Penman-Monteith equation (or as fraction of potential evaporation rate) (Abbott et al., 1986) |
| Snowpack and melt* | Accumulation equation and energy budget melt equation (or degree-day melt equation) (Abbott et al., 1986) |

* Snowpack and melt is not considered in this study.

**Table 1.** Equations of hydrological processes in SHETRAN




| Storm* | Slope (m km-1) | $a_2$ | $b_2$ | Slope (m km-1) | Storm* | $a_3$ | $b_3$ |
|---|---|---|---|---|---|---|---|
| | | $Tp = aL^b$ | | | | | |
| | 14.59 | 0.94 | 1.07 | | 10×1 | 0.59 | 1.14 |
| | 17.51 | 0.86 | 1.08 | | 20×1 | 0.54 | 1.04 |
| | 20.43 | 0.79 | 1.09 | | 30×1 | 0.53 | 0.98 |
| | 23.34 | 0.72 | 1.10 | | 50×1 | 0.38 | 0.99 |
| | 26.26 | 0.64 | 1.12 | | 60×1 | 0.38 | 0.97 |
| | 29.18 | 0.59 | 1.13 | | 80×1 | 0.49 | 0.88 |
| 10×1 | 58.36 | 0.56 | 1.11 | 29.18 | 100×1 | 0.60 | 0.92 |
| | 87.54 | 0.54 | 1.09 | | 200×1 | 2.76 | 0.58 |
| | 116.72 | 0.53 | 1.07 | | 10×2 | 0.66 | 1.00 |
| | Catchment shape | $a_4$ | $b_4$ | | 10×5 | 1.07 | 0.78 |
| | A | 0.59 | 1.14 | | 10×10 | 2.51 | 0.83 |
| | B | 0.44 | 1.23 | | 10×15 | 5.46 | 0.47 |
| | C | 0.63 | 1.17 | | 10×20 | 10.03 | 0.35 |

*Storm is presented by rainfall intensity (mm h$^{-1}$) × duration (h)

**Table 2.** The coefficients of equations of $Tp$ and drainage length





|   | Area (km$^2$) | Drainage length (km) | Slope (m km$^{-1}$) |
|---|---|---|---|
| A | 137 | 12.12 | 29.18 |
| B | 142 | 12.35 | 29.18 |
| C | 140 | 16.97 | 29.18 |


**Table 3.** The catchment properties of the three cloned catchments



| | | $Qp = aL^b$ | | | | $Qp = \dfrac{c}{Tp}$ | |
|---|---|---|---|---|---|---|---|
| Storm* | Slope (m km⁻¹) | $a_5$ | $b_5$ | $c$ | Slope(m km⁻¹) | Storm* | $c$ |
| | 14.59 | 0.45 | -0.86 | 0.74 | | 10×1 | 3.21 |
| | 17.51 | 0.57 | -0.94 | 0.79 | | 20×1 | 3.86 |
| | 20.43 | 0.59 | -0.94 | 0.81 | | 30×1 | 3.99 |
| | 23.34 | 0.61 | -0.94 | 0.83 | | 50×1 | 4.31 |
| | 26.26 | 0.64 | -0.94 | 0.83 | | 60×1 | 4.83 |
| | 29.18 | 0.65 | -0.93 | 0.8 | | 80×1 | 5.22 |
| 10×1 | 58.36 | 0.82 | -0.95 | 0.88 | 29.18 | 10×10 | 7.56 |
| | 87.54 | 0.94 | -0.96 | 0.92 | | 10×20 | 9.04 |
| | 116.72 | 0.98 | -0.94 | 0.9 | | 100×1 | 5.82 |
| | Catchment shape | $a_5$ | $b_5$ | $c$ | | 200×1 | 5.49 |
| | A | 0.65 | -0.93 | 0.8 | | 10×2 | 3.68 |
| | B | 0.67 | -0.98 | 0.8 | | 10×5 | 5.93 |
| | C | 0.99 | -1.03 | 0.99 | | 10×15 | 8.49 |

*Storm is presented by rainfall intensity (mm h⁻¹) × duration (h)

**Table 4. The coefficients in equations of $Qp$**





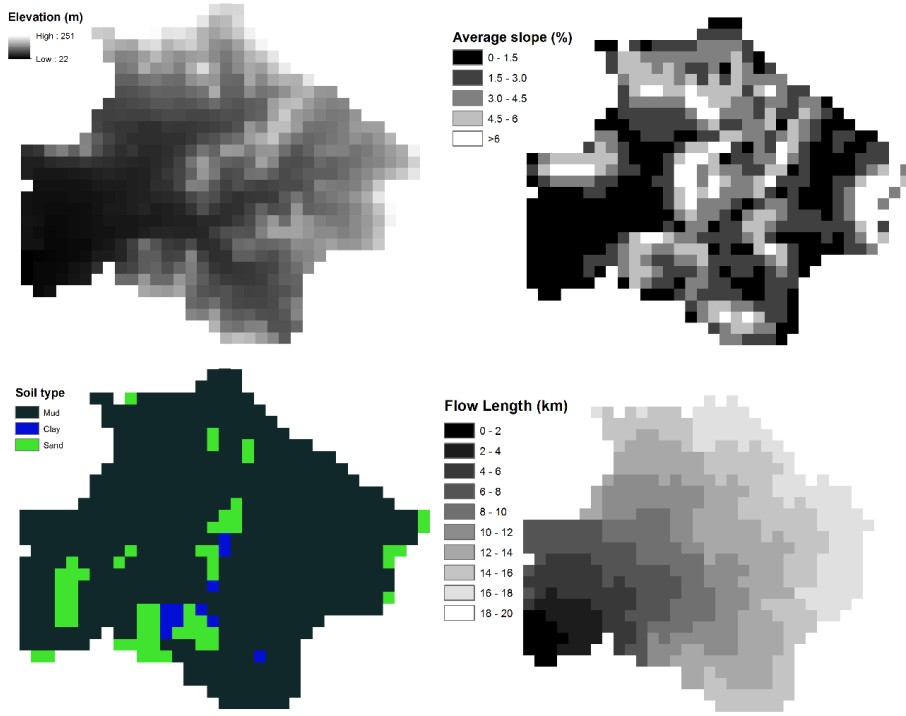


**Figure 1. Spatial data for the baseline model of the Brue catchment (500m grid)**



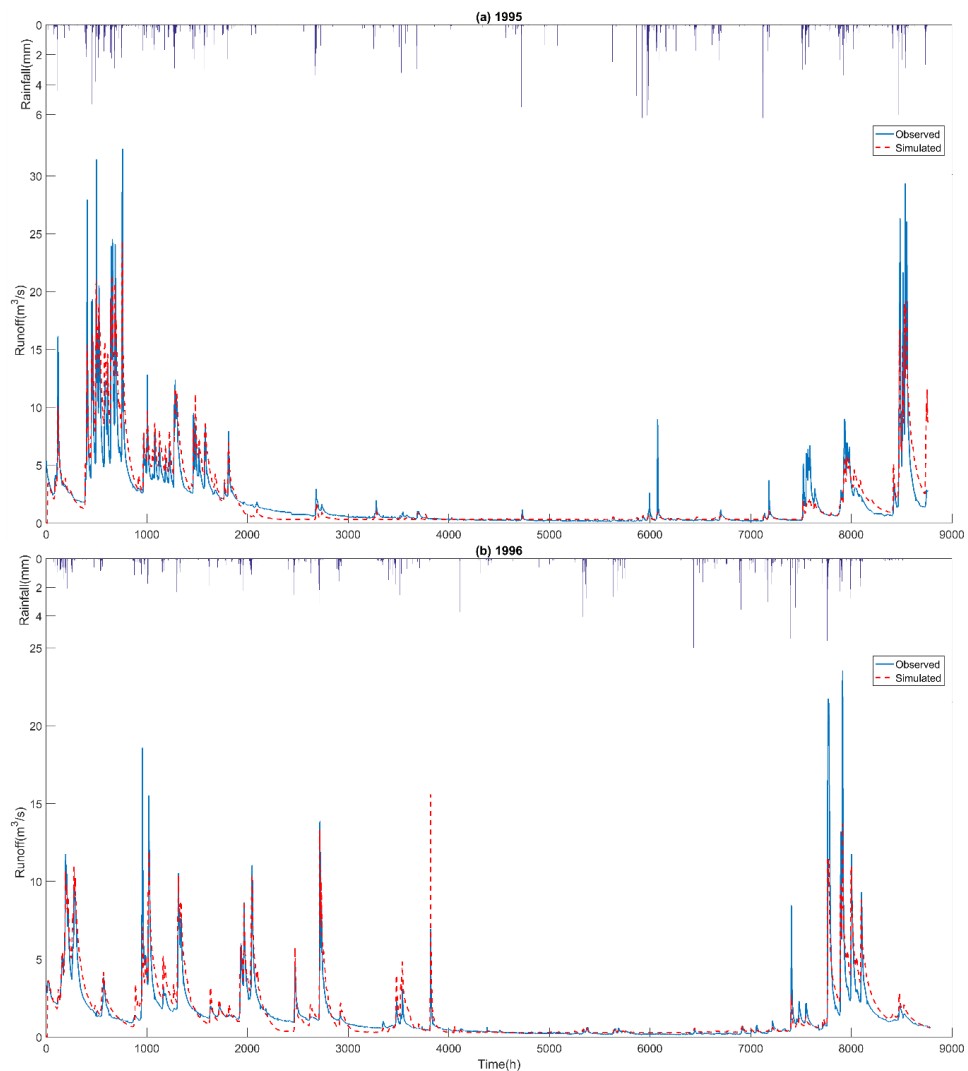


**Figure 2. Hydrographs of the model calibration and validation**





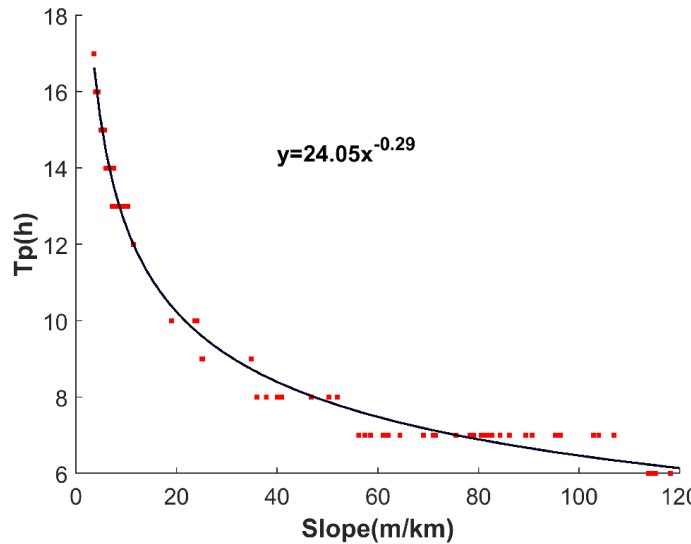


**Figure 3. The relationship between *Tp* and the average slope**





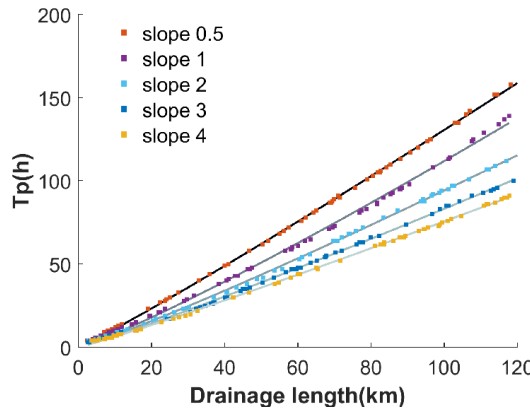


**Figure 4. The relationship between $Tp$ and the drainage length with different slopes**





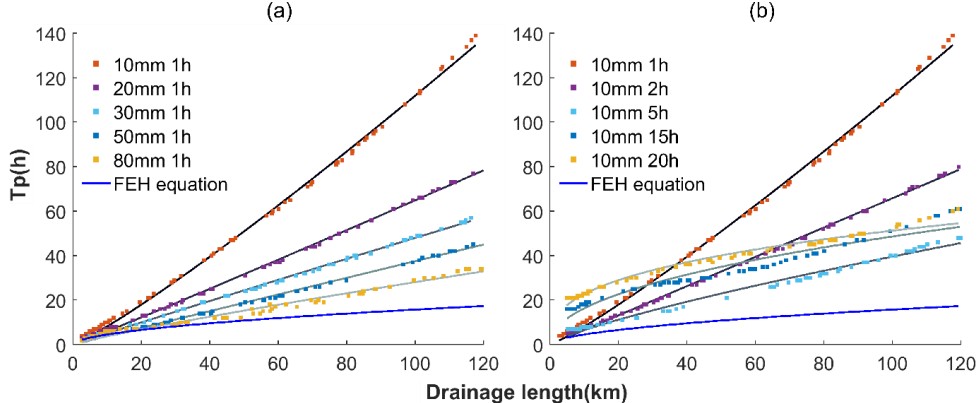


**Figure 5. The relationship between $Tp$ and drainage length in different storms**





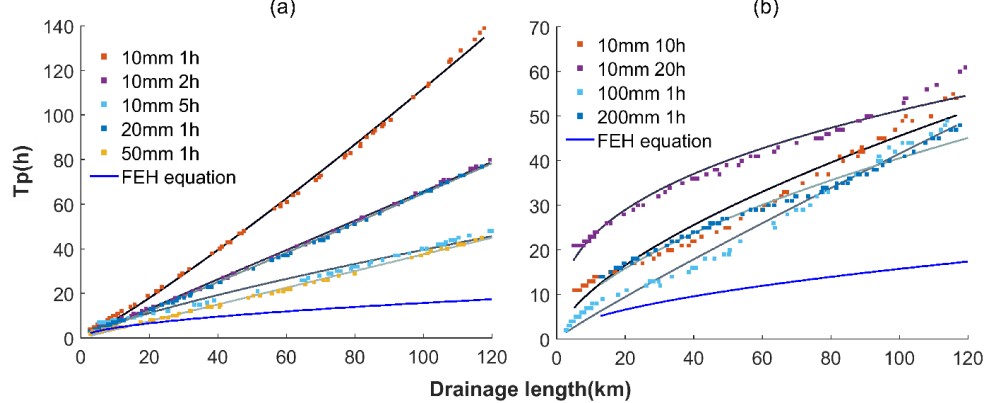


**Figure 6.The relationship between $Tp$ and the drainage length in different storm patterns**





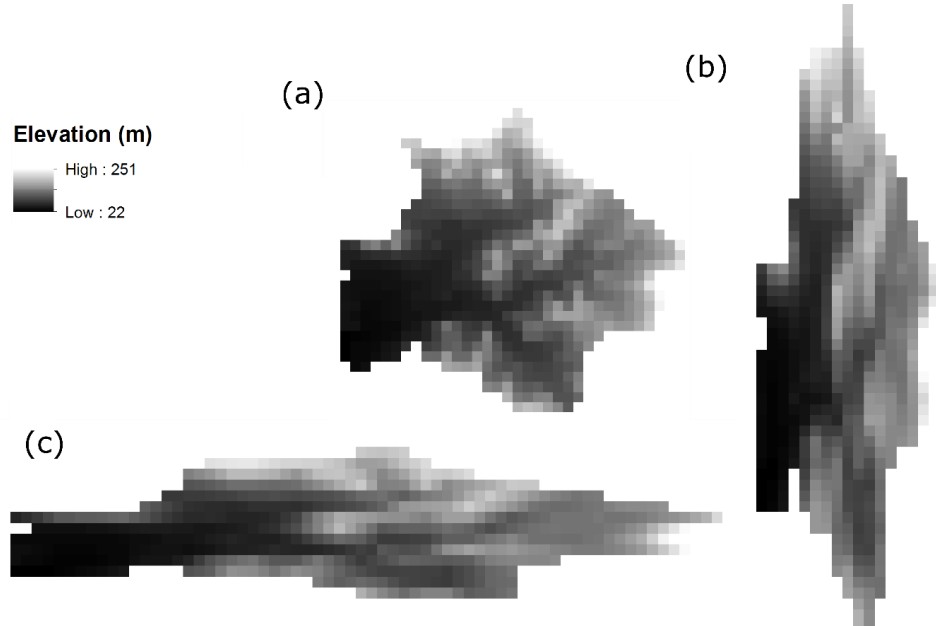


**Figure 7. Three catchments with different shapes**




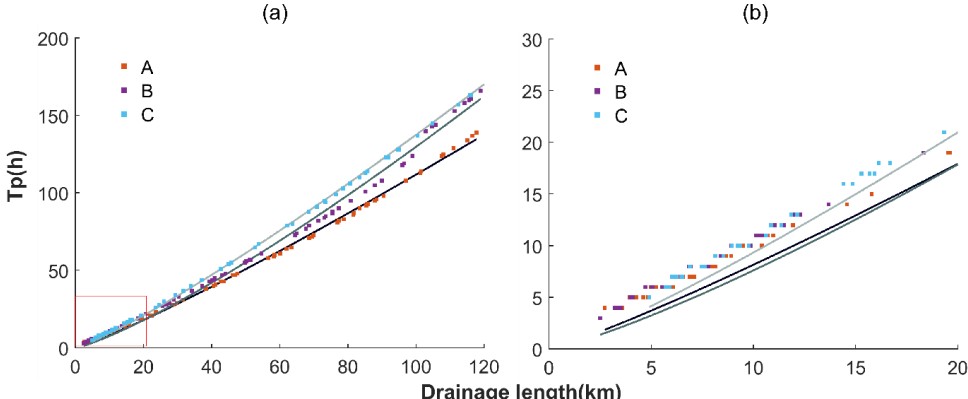


**Figure 8. The relationship between $Tp$ and drainage Length in three catchments with different shapes**





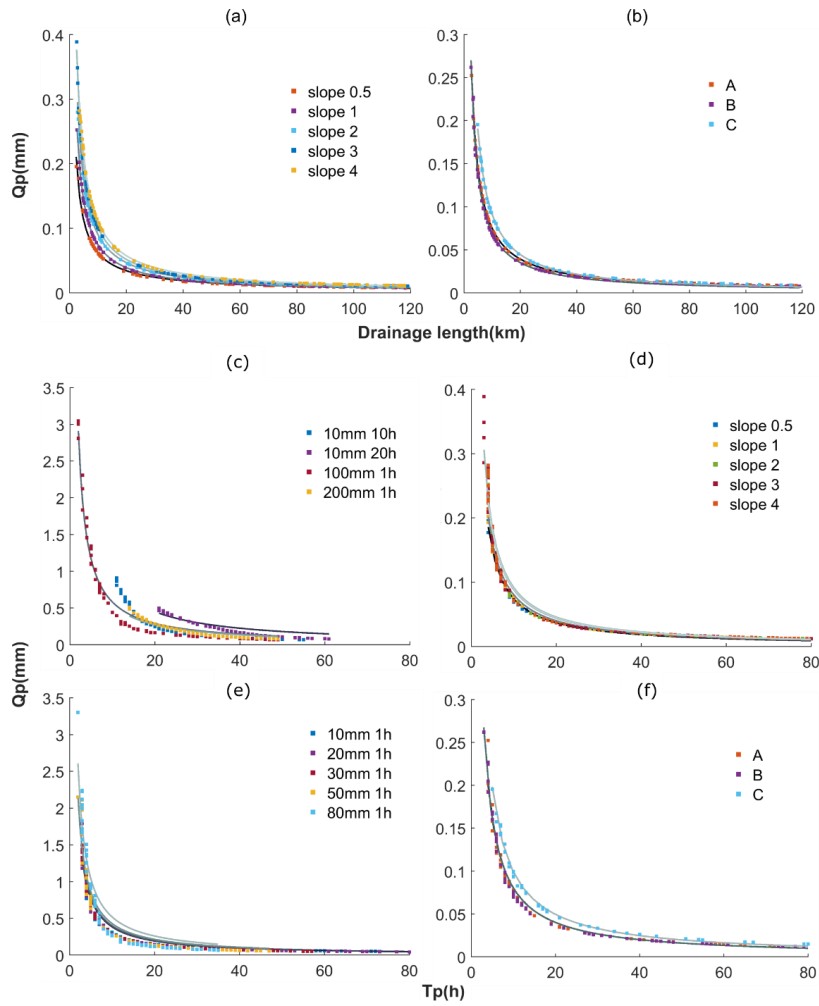


**Figure 9. The relationship between $Qp$ and Drainage length (a)-(b), $Qp$ and $Tp$ (c)-(f) in different slopes and shapes**