# Peer review of "Exploration of virtual catchments approach for runoff predictions of"

_Hydrology and Earth System Sciences, 2017_

## Referee Comment (RC1) · Anonymous Referee #1 · 13 Jun 2017

The manuscript explores unit hydrograph properties through virtual catchment analysis and a distributed rainfall-runoff model. The approach is interesting and potentially useful, however there some basic choices made by the authors that limit the usefulness of the proposed study.

Reading the paper it is seems that the empirical FEH Unit Hydrograph is the perfect solution for ungauged basin, and that, this is the approach to be preferred, so that it is worth to investigate on its parameter. However, yet in the Introduction, authors list a series of possible alternatives that, paradoxically, seems to solve the problem stated as aim of the paper.

In my view, the WFIUH approach is already able to adsorb from the DEM all the geomorphological properties of the specific analyzed ungauged watershed. The analyst

does not need to transfer from an other similar watershed geomorphological information in order to build it. This is the great value of the WFIUH approach. There are only one (or two) parameter to be assigned for its definition but they are more related to the kinematic watershed properties than to the geomorphological ones.

If the aim focuses strictly on the FEH approach, as it seems, maybe it could be appropriate to submit the paper (with a less general title) to a more applied Journal.

Say that, the idea to use a virtual watershed could be interesting as well for investigation the role of WFIUH parameters, also if it is well known that at the end, in ungauged basin, the net rainfall estimation step is much more influencing than the IUH definition.

Concerning the approach proposed by the authors in order to simulated similar catchments with different slope or other attributes, I am little bit skeptical on just multiplying the elevation (or the attribute) by a factor. For sure it would influence the geomorphological properties. It would be more interesting trying to simulate topographic surface with a more advanced methods that can control watershed properties (i.e. Grimaldi S, Teles V, Bras RL (2005). Preserving first and second moments of the slope area relationship during the interpolation of digital elevation models. Advances in Water Resources, vol. 28, p. 583-588, ISSN: 0309-1708.)

---

## Author Comment (AC1) · 14 Jun 2017

Many thanks for the useful and constructive comments. We do apologise that the original manuscript is not sufficiently clear in some statements, which are clarified in the following.

The main purpose of this study is to investigate the possibility of using the virtual catchment approach on runoff generation influenced by catchment geomorphology with potential applications in ungauged catchments. Rather than focusing on the parameters in UH equations, this study concerns more about the reliability of runoff generation by the approach, i.e. virtual catchment.

(1) UH is a useful tool for ungauged catchment and catchment morphology is crucial in

runoff prediction. Although there have been researches on UH combined with geomorphology characteristics, limitations exist in the current approaches. More discussions will be added on this in the revised introduction.

(2) WFIUH has been verified as a powerful approach in ungauged catchments. However, it should be noted that not only elevation data is crucial to runoff generation, but also land use, soil types and storm patterns (demonstrated in the manuscript) are significant as well, therefore, WFIUH is limited in dealing with more complex scenarios.

(3) The FEH equation was developed by the Institute of Hydrology in the UK and widely used in practice. The equation has been verified as a useful tool in practice, therefore, by comparing the results from the virtual catchment approach with the FEH equation, it is able to test whether the proposed approach is capable of producing hydrologically meaningful results. According to the outcomes, the trends of catchment geomorphology on runoff generation are similar while influenced by more indicators than that used in the FEH equation.

(4) The future of the virtual catchment approach is more than simulating runoff from catchments with similar characteristics, but applying it to ungauged catchments with varied conditions using more advanced methods. This is a first attempt to explore and promote the approach to ungauged catchments. More advanced catchment transformation will be generated in the future for broader applications.

(5) Thanks for the references which will be added in the revised manuscript.

To improve the paper's readability, we will include these clarifications into the revised manuscript.

---

## Referee Comment (RC2) · Anonymous Referee #2 · 7 Aug 2017

The authors have provided an interesting use of virtual catchments to explore the sensitivity of the unit hydrograph to several geomorphological features. The project is well-conceived and provides an interesting tool for continued analysis. I would like to see the authors strengthen their statistical presentation, address a one concern in the logic of their argument and more-fully describe the potential for ungauged applications.

Firstly, I'd like the authors to consider the statistical significance of their results. Several power functions are fit to different curves. Each time, the fitted coefficients are contrasted with the coefficients implied by the Flood Estimation Handbook. The authors fail to provide any hypothesis test on the strength of their evidence. For example, in line 161, the coefficient is fitted as -0.29, contrasting with the implied -0.35. While the original value (-0.35) probably contains some degree of uncertainty, the fitted value

(-0.29) is certainly uncertain. Depending on how these regressions were fit (I'm assuming ordinary least squares, though this should appear in the methods section), it is possible to execute a hypothesis test to determine if the resulting fitted value is significantly different from the expected value from the Flood Estimation Handbook. (This raises a minor point, again on line 161, where "significant" is used in a qualitative sense. Given the statistical meaning of the word in this context, I'd suggest avoiding value judgements or finding a synonym.)

In addition to considering the significance of the results, I would ask the authors to consider their fitted equations a bit more closely. The authors fail to provide any assessment of goodness of fit to portray the accuracy of their estimates. Furthermore, do these regressions meet the assumptions of their approach? If we are talking about ordinary least-squares fitting, then we need to understand the performance of the residuals before we can begin to trust coefficient estimates. For example, consider figure 5: The residuals of the 10mm/h storm appear highly non-Normal. They display uncaptured, systematic curvature. This certainly raises concerns about the validity of the coefficients. While I understand that the intent is to contrast with the equation from the Flood Estimation Handbook, this limitation should be acknowledged and the implications thereof discussed.

Moving on, I would like the authors to provide some further consideration of the logic of their argument that virtual catchments allow us to assess underlying conditions across diverse physical conditions. It seems to me that the use of a hydrologic model forced through a set of virtual catchments is a stronger reflection of the underlying model rather than reality. Of course, with a well-performing model, this point is moot: The model approaches reality, and, therefore, the behavior of the forced model is the behavior of reality. However, I think it should be acknowledged that, without a thorough assessment of all aspects of model performance, we can never be certain that they curves we identify (as in Figure 5 and elsewhere) are a reflection of underlying physical properties (reality) rather than a product of the model's conceptualization of reality.

[Figure]

While this may not discredit the authors' findings, I think it is an important limitation to discuss.

Additionally, I would like the authors to consider expanding the discussion of ungauged applications. The authors claim that the use of virtual catchments allows models to be applied in ungauged regions, avoiding the need to transfer parameters of the unit hydrograph. However, the authors then proceed, in their first step, to calibrate their model the site of interest. The fact that a model must first be calibrated to a particular site makes ungauged application particularly difficult. Now, it may be that the authors are instead arguing that a model calibrated to an arbitrary catchment can then be forced with virtual catchments similar to the ungauged region of interest, but this is not immediately clear from the discussion. Furthermore, it seems somewhat ill-advised to take a model calibrated to one setting and force it with artificial settings. The mere fact that calibration is required implies that the underlying model is not a perfect representation of reality: why, therefore, would we expect it to behave across a range of virtual catchments when re-calibration would be required across a diverse range of observed catchments? Again, I am not seeking to discredit the authors' argument, but am merely hoping my concerns might foster more discussion in the manuscript.

Before closing, let me mention some minor concerns: I would like to see the authors provide a bit more description in their section on methodology. For one, it would be good to describe how the authors plan to apply the concept of virtual catchments. For the uninitiated, it is not clear what is meant by a virtual catchment until it is described in the results section. Additionally, the authors acknowledge the limitations of the Nash-Sutcliffe model efficiency in section 3.1, but I would ask them to explore other metrics like the efficiency of the logarithms or square roots, or still other metrics. As the performance of the underlying model is essential, as I argued above, I think it worthwhile to further demonstrate the accuracy and precision of the model. Finally, the authors should provide the methodology that was used to estimate the curves throughout the results section.

Finally, I sincerely thank the authors for their work and their well-written manuscript. It was easy to read, which made it easy to consider its technical merit. I look forward to their considered response and hope to have ignited some useful thought and discussion. If you would like any clarification on my comments, I strongly encourage you to reach out to me.

**Possible typographical errors: (not exhaustive, just ones I noticed)**

Line 62: "...especially in because small catchments ..." seems to have an extra word.

Line 118: "patterns" should not be plural

Line 177: I think you mean "fluctuation"

Line 185: You refer to similar trends, but you should probably note that these are similar in sign only. Similarly, you refer to "vast" differences. That is subjective, especially without any formal consideration of significance.

Line 232: Should "The analysis between Qp and L..." be "The analysis between Qp and [Tp]..."? That would seem to match the figure and heading.

Line 236: You refer to significance here, but it appears to have no statistical support. A synonym might be more appropriate.

---

## Author Comment (AC2) · 25 Aug 2017

We appreciate the helpful and constructive comments and the responses to the comments are as follows:

(1) The regressions were fitted with ordinary least squares, which will be added in the methodology section in the revised manuscript. The fitted curve in Figure 3 and line 161 was derived from hourly data to be consistent with the Flood Estimation Handbook (FEH), which is possible to bring some uncertainty. An example is shown here to demonstrate the difference between the fitted curve and FEH equation in line 161. Using the average slope in the Brue catchment, which is 29.12 m/km, Tp is 9.04 h from the fitted curve and is 5.38 h from the FEH equation, with the relative error of 40.4%,

[Figure]

which indicates the difference is significant. More detailed analysis on the difference about all the fitted curves will be discussed in the revised manuscript.

(2) As demonstrated in the comments, the curves for 15 h and 20 h with 10 mm/h in Figure 5(b) did not capture the trends well especially when the drainage length is large. Further details of the curves including R2 and residuals, and more analysis on the limitations of curves will be further discussed in the revised manuscript.

(3) Many thanks for the useful suggestions and we agree that there are still gaps between the hydrologic model and the reality. As the hydrologic model is a simplification of the real system, it is impossible for the model to exactly represent the reality. In this study, the model is able to capture the main hydrologic characteristics of the catchment, which is useful in studying the catchment geomorphology and runoff production. With more available data and a wider exploration of the virtual catchment method, more understanding on the gaps between the model and the reality will be gained in order to reduce them in the future. More discussions about the possible limitations will be illustrated in the Discussion section.

(4) The baseline model was calibrated because the required data of the model were not all available such as land use and soil depth. The purpose of the study is to investigate the reliability of using the virtual catchment approach to simulate the catchment response to the changing geomorphology. To apply the approach to ungauged catchments, the geomorphological characteristics, e.g. slope, drainage length, area, etc., of the baseline model are to be modified according to the ungauged catchment, which results in a new representation for the ungauged catchment. As the model is fully physical-based and demonstrated its reliability in runoff generation, the new model can be treated as a reasonable representation of the ungauged catchment. Therefore, the runoff generated from the target model is a simulation of the ungauged catchment. The application of the virtual catchment approach will be further explored with more available catchments data. We will add this discussion in the revised manuscript.

(5) The descriptions of the virtual catchment approach will be added in the methodology section. A more comprehensive description of the fitted curves will be provided in the revised manuscript.

(6) Line 62: Yes, 'because' is an extra word and will be removed. Apologies for the mistake.

Line 118: Yes, it will be changed to 'pattern'.

Line 177: Yes, it should be 'fluctuation'.

Line 185: Yes, as discussed before, a statistical test will be performed to define the significance of the difference to obtain more precise discussions.

Line 232: As the relationship between Qp and L was shown in Figure 9 (a) and (b), it was discussed before the discussion of the relationship between Qp and Tp. Apologies for the insufficiently clear statements. The section title will be changed to 'Relationship between Qp with L and Tp'.

Line 236: Yes, the sentence will be re-organized as 'The effect of geomorphological characteristics on peak flows is more significant for small catchments as Qp varies more significantly with different slopes in the small catchments (with shorter drainage length)'.

A thorough check the spelling and grammar will be carried out in the revised manuscript.